# Representation of Parkinson's disease and atypical Parkinson's syndromes in the Czech Republic—A nationwide retrospective study

Jiří Bůřil[1☯¤*], Petra Bůřilová[2,3☯], Andrea Pokorná[2,3‡], Ingrid Kováčová[3], Marek Baláž[1¤‡]

1 I[st] Department of Neurology, Faculty of Medicine, Masaryk University, Brno, Czech Republic,
2 Department of Nursing and Midwifery, Faculty of Medicine, Masaryk University, Brno, Czech Republic,
3 The Institute of Health Information and Statistics of the Czech Republic, Prague, Czech Republic

☯ These authors contributed equally to this work.
¤ Current address: Ist Department of Neurology, St. Anne´s University Hospital, Brno, Czech Republic
‡ These authors also contributed equally to this work.
* Jirka210312@gmail.com

## Abstract

### Background

Parkinson's disease is a progressive neurodegenerative disease which causes health problem that affects more patients in the past few years. To be able to offer appropriate care, epidemiological analyses are crucial at the national level and its comparison with the international situation.

### Aim

The demographic description of reported patients with parkinsonism (including Parkinson's disease and atypical parkinsonian syndromes) according to the International Classification of Diseases (ICD-10) from the national health registries.

### Methods

Retrospective analysis of data available from the National Health Information System–NHIS and the National Registry of Reimbursed Health Services (NRRHS). Analyzed epidemiological data are intending to determine the regional and specific prevalence of Parkinsonism in the Czech Republic. The International Classification of Diseases diagnoses (ICD-10) of G20 (Parkinson's disease—PD) and G23.1, G23.2, G23.3 (other degenerative disorders of basal ganglia), and G31.8 (another degenerative disease of basal ganglia) from the period of 2012 to 2018 were included into the analysis.

### Results

We identified 78 453 unique patients from national registries in the period 2012 to 2018. Diagnoses of G20, G23.1, G23.2, and G31.8 were registered as the principal diagnoses in 76.6% of all individual patients.

**Data Availability Statement:** There are legal restrictions on sharing data publicly. All data from the national registries must be kept safely and authors are working with the anonymised data.

Data access requests can be sent to the head of the department of data Analyses team from the Institute of Health Information and Statistics of the Czech Republic - Dr. Jiří Jarkovský (email: Jiri. Jarkovsky@uzis.cz).

**Funding:** The authors received no specific funding for this work.

**Competing interests:** The authors have declared that no competing interests exist.

## Conclusion

We have found a growing number of patients coded with ICD-10 of dg. G20, G23.1, G23.2, G23.3, or G31.8 (N = 27 891 in 2012, and N = 30 612 in 2018). We have proven regional differences in the prevalence of Parkinson´s diagnoses. Therefore we assume most likely also differences in the care of patients with PD based on the availability of specialty care centers.

## Introduction

Parkinson's disease (PD) is a progressive neurodegenerative disease that affects mainly the function of basal ganglia. Common PD motor signs include bradykinesia, rigidity, tremor, and postural instability. Incidence and prevalence of PD appear to increase over the recent years [1]. PD signs tend to progress over the period of several years from early non-motor symptoms to fully developed parkinsonian syndrome as described by Braak pathological stages [2]. Motor signs, e.g., parkinsonian syndrome, can be controlled by dopaminergic treatment, especially in the early stages of the disease. Advanced stage of PD—accompanied by late/advanced motor symptoms and fluctuations of motor state—usually occurs within 5 to 7 years since the onset of the disease [3]. Therapeutic options become more complex with the advancing PD stage [4].

The prevalence of the disease ranges from 1 to 3 per 1000 in unselected populations, and it affects 1% of the population above 60 years [5]. According to a recent study, methodological differences between studies make a direct comparison of prevalence estimates difficult [6, 7]. Global Burden of Disease study points to estimates of approximately 13 million people treated for PD by 2040 [8]. Last available data from IHIS CR (Institute for Health Information and Statistics of the Czech Republic–

Ústav zdravotnických informací a statistiky České republiky) from 2012 report 26 680 patients [9]. The population of the Czech Republic in 2012 reached approximately 10 516 000 people. The prevalence at that time point was 2.53 per 1000 inhabitants. Young-onset PD patients with diagnosis established before the age of 40, comprised 15% [10]. Other papers have reported different prevalence of data [11, 12]. Available data show the average age of PD diagnosis between 50 and 60 years. Therefore, it is clear that the disease affects people of productive age. PD prevalence grows with increasing age, and it will further increase due to an increase in longevity. We decided to perform a retrospective analysis of national health statistic data to elucidate a number of patients treated for PD in the Czech Republic as we are aware that the valid data on prevalence may be necessary for both healthcare providers and health and social care payers, as well as for patient groups.

### Aim of the analyses

The demographic description of reported patients with parkinsonism (including Parkinson's disease and atypical parkinsonian syndromes) according to the International Classification of Diseases (ICD-10) from the national health registries.

## Materials and methods

We performed a retrospective analysis of data available from NHIS (National Health Information System), respectively NRRHS (National Registry of Reimbursed Health Services) with respect to the STROND checklist. In both registries, the data from health care providers about

the type and amount of care are collected. We analyzed epidemiological data intending to determine the regional and specific prevalence of Parkinsonism in the Czech Republic. The International Classification of Diseases (ICD-10) diagnoses of G20 (Parkinson's disease) and G23.1, G23.2, G23.3 (other degenerative disorders of basal ganglia), and G31.8 (another degenerative disease of basal ganglia) from the period of 2012 to 2018 were included into the analysis. Atypical parkinsonian syndromes were included due to overlap of clinical signs, at least during the initial stages of Parkinsonism. We selected the period till 2018, as the last previous data on national prevalence were published in 2018. The diagnosis G23.3 (according to ICD-10) was not found in the analyzed data, so we excluded this diagnose from our further evaluation. The standardized annual prevalence of the patients with parkinsonism (including Parkinson's disease and atypical parkinsonian syndromes) has been counted. The analyzes of age and gender distribution, age in the date of diagnosis, and regional distribution of patients within the Czech Republic districts in the analyzed sample.

### Patient and public involvement

No patients were involved in the design of the study. For the research purposes analyses, there is not special ethical approval needed according to the Czech law if they are analyzing by the Institute of Health Information and Statistics and not by the third party.

### Results

We identified 205 490 records of patients with PD from national registries in the period from 2012 to 2018 (averaging 29 000 patients per year, Fig 1), which means 78 453 unique patients in total (each patient could be recorded just one time during the year, but several times during the whole period from 2012 to 2018). We have found a growing number of records coded with ICD-10 of dg. G20, G23.1, G23.2, or G31.8 (N = 27 891 in 2012, and N = 30 612 in 2018). The diagnosis G23.3 (according to ICD-10) was not found in the analyzed data, so we excluded this diagnose from our further evaluation. Diagnoses of G20, G23.1, G23.2, a G31.8 (International Classification of Diseases–ICD-10) were registered as the principal diagnoses in 76.6% of all individual patients. After extrapolation to the overall population, a standardized annual prevalence reached 276 per 100 000 inhabitants. After the age and gender distribution was taken into account, a higher incidence was found in the age group of over 65 years, 48% of patients were men, 52% women. The average patient age was lower in males (73 years) than female patients (75 years) (Fig 2). Most frequent age at diagnosis (i.e., age at which a diagnosis was registered with diagnosis in the health care system) was in the range of 75–79 years. We determined the number of patients with these diagnoses concerning regional distribution within districts of the Czech Republic (Fig 3).

### Discussion

Based on the results of our analysis, we identified approximately 29 000 records of patients per year treated for Parkinsonism in the Czech Republic. Only 4.2% (from the whole period 2012–2018) of these patients were followed under ICD-10 diagnostic codes of atypical Parkinsonism (G23.1, G23.2, G23.3, G31.8). According to available published data, the reported prevalence of atypical Parkinsonism is 0.4% (400 cases per 100 000 populations) in dementia with Lewy bodies (G31.8), multisystem atrophy with dominant parkinsonian features (MSA-P, G23.2), and progressive supranuclear palsy (PSP, G23.1) in 5 to 10 cases per 100 000 of the general population, increasing to 7 to 8 per 100 000 inhabitants in elderly [13]. The ratio of the parkinsonian subtype of MSA to the cerebellar subtype is 2:1 to 4:1 in 66 countries [14, 15]. Diagnosis code G23.3 of multiple system atrophy—cerebellar type (a rare disorder with an estimated

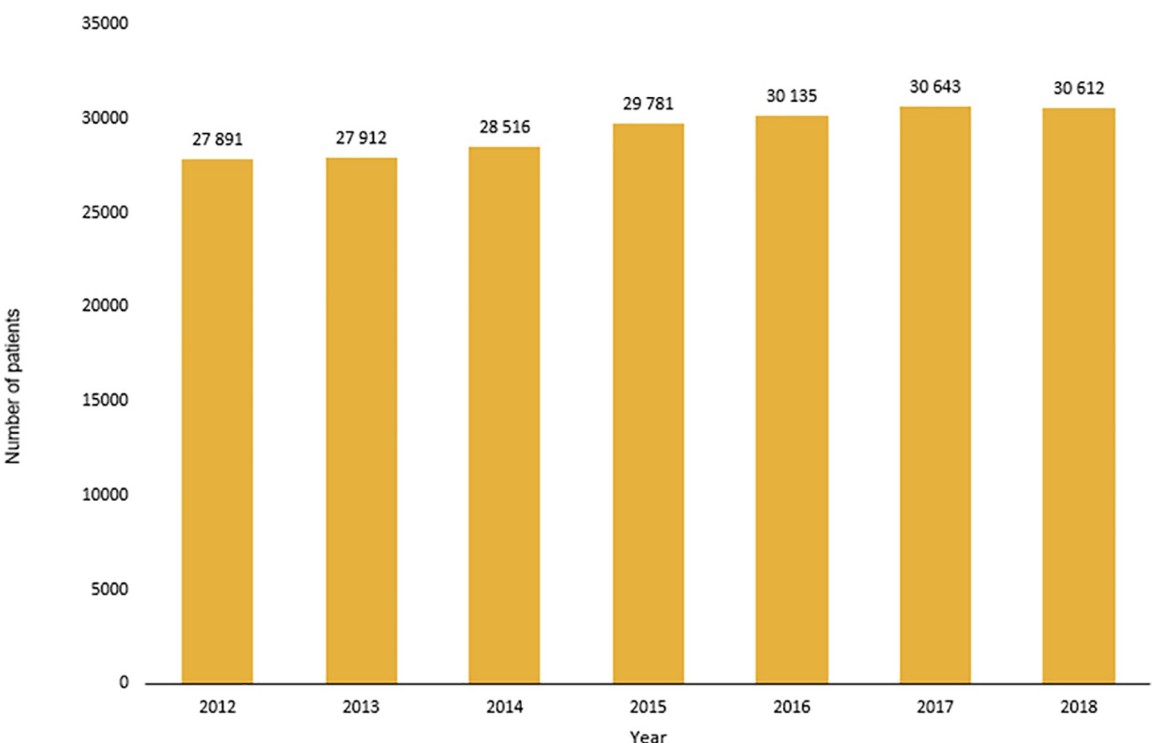

**Fig 1. Number of records patients with diagnosis G20, G23.1, G23.2 or G31.8 in individual years.**

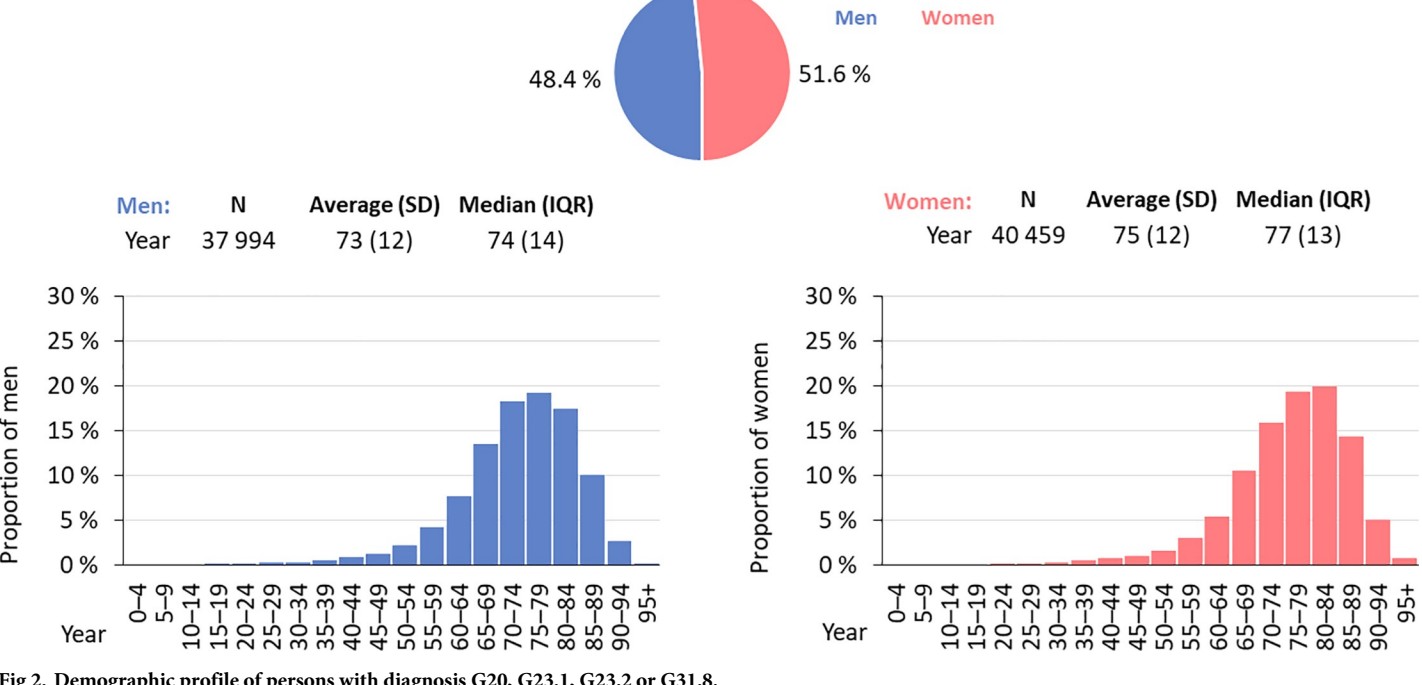

**Fig 2. Demographic profile of persons with diagnosis G20, G23.1, G23.2 or G31.8.**

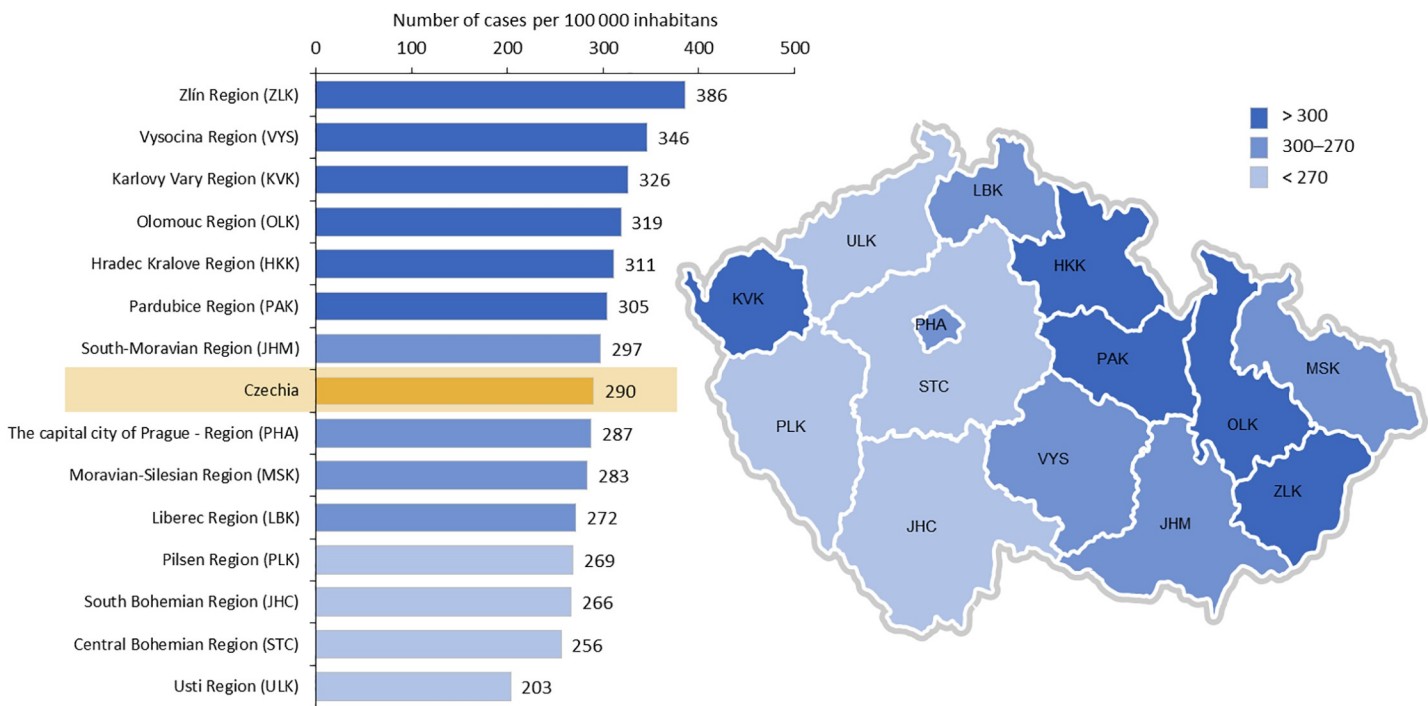

**Fig 3. Number of patients with diagnosis G20, G23.1, G23.2 or G31.8 by residence in relation to the population of the region in the Czech Republic.**

average prevalence of 0.6 to 0.7 per 100 000 inhabitants [16]) was not present in the Czech national health registries. The possible reason is, that there were no incidence of patients with dg. G23.3 which could relate to the problem that the clinicians are unable to diagnose it appropriately. Also published literature is confirming that the dg. G23.3 is not reported, the commonly used codes are 23.1, 23.2 and 23.9 [17]. Another possibility is that the clinicians do not show dg. multisystem atrophies correctly under the code G 23.3, but under the general one, for example G31.8. Our analysis has shown an average prevalence of 9.8 per 100 000 of code G31.8, 1.2 per 100 000 of code G23.1, and 0.7 per 100 000 of code G23.2 (from the whole period 2012–2018). We did not report cases of corticobasal degeneration. Those are reserved for the neuropathological diagnosis. However, it can be estimated that at least a certain share of patients coded as G20 (PD) was, in fact, patients with atypical parkinsonian syndromes. Therefore, atypical parkinsonian syndromes may be underdiagnosed. However, they should not be underestimated in the differential diagnosis of Parkinsonism [13].

According to data available from health registries (NHIS), we have shown that most patients diagnosed and treated with PD were in the age group of 70–84 years (50.3% male and 49.7% women). Age at the time of PD diagnosis was 70–84 years, which differs from available EU data [16, 18]. European data sets show age at the time of diagnosis lower by 10–15 years. We speculate that inadequate diagnosis or late registering of a patient under the correct ICD code may be the cause of a later diagnosis age, found at our registries. We cannot assume that the diagnosis and treatment of a patient with Parkinsonism occur late. Diagnostic accuracy may differ concerning disease duration (lower during the first contact of a patient with early parkinsonian symptoms), age of a patient (possibility of later diagnosis in younger patients), the experience of a physician, and understanding of PD. Diagnostic mistakes may be caused by overlapping symptoms in the early stages of Parkinsonism and the similarity of other disorders (essential tremor, dystonic tremor). The current diagnosis of PD is aided by established

clinical diagnostic criteria [19, 20]. Some patients may lack early access to a neurologist specialized in movement disorders, perhaps also due to the uneven geographic distribution of centers focused on movement disorders. There are two centers in the eastern part of the country—(Moravia and Silesia with almost 5 million inhabitants) and a single center in the western part (Bohemia, with approximately 5.5 million inhabitants). Another possible cause of potentially late diagnosis may be due to limited healthcare awareness and underestimation of disease signs, as was shown in 59.4% of the Czech population by a survey related to the preparation of the program "Health 2020" in the Czech Republic [21]. Various genetic and social factors may influence the regional distribution of patients with Parkinsonism, as was already reported in the eastern part of our country [22]. We did not have the possibility to deeper analysed the data about recorded patients (gender and age) and because of this we have look at the records based on the place where the care was provided. The highest rate of PD diagnosis was found in districts of Zlin Region (386 cases per 100 000 inhabitants), Vysočina Region (346 cases per 100 000 inhabitants), and Olomouc Region (319 cases per 100 000 inhabitants). The exact location of the residence was not possible to determine in 247 cases analyzed for 2018.

We are aware of several limitations of this study. Both unintentional and intentional coder errors, such as misspecification, and up-coding, as potential sources of errors, were described [23]. As we have limited knowledge about the exact diagnostic coding by individual physicians and in various hospitals over the country, we cannot determine the level of coding mistakes that may have occurred, similarly as was reported in a recent study on diagnostic accuracy in Parkinsonism [24, 25].

The importance of the availability of movement disorder centers is underscored by data showing that nearly half (47%) of PD diagnoses are incorrect when performed in the primary care setting. Specialists without expertise in movement disorders have an error rate of approximately 25%, while movement disorder specialists made mistakes in only 6% to 8% of cases [26, 27].

## Conclusion

According to available epidemiological data from the Czech health care registries, we were able to perform an analysis of population data concerning ICD-10 codes coding Parkinsonism and related disorders. We also analyzed the regional prevalence of these diagnoses in respective regions of the Czech Republic. We have found a growing number of records coded with ICD-10 of dg. G20, G23.1, G23.2, G23.3, or G31.8 (N = 27 891 in 2012, and N = 30 612 in 2018). We have proven regional differences in the prevalence of diagnoses. Therefore we assume most likely also differences in the care of patients with PD based on the availability of specialty care centers. The big issue for the future in the Czech Republic is the implementation of the standardized processes of care based on nationally accepted evidence-based guidelines.

## Author Contributions

**Conceptualization:** Jiří Búřil, Petra Búřilová, Andrea Pokorná, Ingrid Kováčová, Marek Baláž.

**Data curation:** Petra Búřilová, Andrea Pokorná, Ingrid Kováčová.

**Formal analysis:** Ingrid Kováčová.

**Methodology:** Jiří Búřil, Petra Búřilová.

**Supervision:** Andrea Pokorná, Marek Baláž.

**Validation:** Jiří Búřil, Petra Búřilová, Andrea Pokorná.

**Visualization:** Petra Búřilová.

**Writing – original draft:** Jiří Búřil.

**Writing – review & editing:** Jiří Búřil, Andrea Pokorná, Marek Baláž.

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
