## [Decision Letter · Decision Letter 0]

27 Nov 2020

PONE-D-20-31004

Representation of Parkinson's disease and atypical Parkinson's syndromes in the Czech Republic - A nationwide retrospective study

PLOS ONE

Dear Dr. Búřil,

Thank you for submitting your manuscript to PLOS ONE. After careful consideration, we feel that it has merit but does not fully meet PLOS ONE’s publication criteria as it currently stands. Therefore, we invite you to submit a revised version of the manuscript that addresses the points raised during the review process.

We look forward to receiving your revised manuscript.

Kind regards,

Weidong Le

Academic Editor

PLOS ONE

Journal Requirements:

2.Thank you for stating the following financial disclosure:

 [The funders had no role in study design, data collection and analysis, decision to publish, or preparation of the manuscript.].

3.We note that you have indicated that data from this study are available upon request. PLOS only allows data to be available upon request if there are legal or ethical restrictions on sharing data publicly. For more information on unacceptable data access restrictions, please see http://journals.plos.org/plosone/s/data-availability#loc-unacceptable-data-access-restrictions.

4.We note that [Figure(s) 3] in your submission contain map images which may be copyrighted. All PLOS content is published under the Creative Commons Attribution License (CC BY 4.0), which means that the manuscript, images, and Supporting Information files will be freely available online, and any third party is permitted to access, download, copy, distribute, and use these materials in any way, even commercially, with proper attribution. For these reasons, we cannot publish previously copyrighted maps or satellite images created using proprietary data, such as Google software (Google Maps, Street View, and Earth). For more information, see our copyright guidelines: http://journals.plos.org/plosone/s/licenses-and-copyright.

1.    You may seek permission from the original copyright holder of Figure(s) [3] to publish the content specifically under the CC BY 4.0 license. 

Reviewers' comments:

Reviewer's Responses to Questions

**Comments to the Author**

1. Is the manuscript technically sound, and do the data support the conclusions?

Reviewer #1: Yes

Reviewer #2: Yes

2. Has the statistical analysis been performed appropriately and rigorously? 

Reviewer #1: I Don't Know

Reviewer #2: I Don't Know

3. Have the authors made all data underlying the findings in their manuscript fully available?

Reviewer #1: Yes

Reviewer #2: Yes

4. Is the manuscript presented in an intelligible fashion and written in standard English?

Reviewer #1: Yes

Reviewer #2: Yes

5. Review Comments to the Author

Reviewer #1: In this manuscript, the authors reported a retrospective analysis of data available from the National Health Information System – NHIS and the National Registry of Reimbursed Health Services (NRRHS) with respect to the STROND checklist to determine the prevalence and other epidemiological features of Parkinsonism in the Czech Republic. Data on parkinsonian syndromes (Parkinson's disease and mainly Atypical Parkinsonian syndromes) in different Eastern European countries are scarce and this study could be valuable by depicting epidemiological features of these diseases in understudied populations. However, some changes should be reconsidered.

Abstract:

The aim of the study is not well stated.

The results are not well defined (what do the authors mean by "We identified 78 453 patients? this figures stands for what exactly?

The conclusion part of the abstract included data not previously presented in the results. This should be addressed.

Introduction:

The aim at the end of the introduction part is not clearly stated " to elucidate a number of patients treated for PD in the Czech Republic". The authors should be more specific. They stated later in the Materials and Methods part: " We analyzed epidemiological data intending to determine the regional and specific prevalence of Parkinsonism in the Czech Republic." Objectives should be mentioned all together at the end of the Introduction part.

Materials and Methods:

- The authors should define the exact meaning of the different codes used in this part : " The International Classification of Diseases 33 (ICD-10) diagnoses of G20 (Parkinson’s disease) and G23.1, G23.2, G23.3 (other degenerative disorders of basal ganglia), and G31.8 (another degenerative disease of basal ganglia)"

-How do the authors explain the fact the diagnosis G23.3 (according to ICD-10) was not found in the analyzed data? and why did they decide to excluded this diagnosis from further evaluation? This is a result they could discuss later on in the discussion part.

-What do the authors mean by " regional" and "specific" prevalence of Parkinsonism. This should be better specified in the Methods part.

-What do the authors mean by: " Atypical parkinsonian syndromes were 36 included due to overlap of clinical signs, at least during the initial stages of Parkinsonism."? How does this affect their methodology? this should be better explained.

-The authors should precise all the data analyzed later in the results in the methodology part.

Results:

-What do the authors mean by "we identified 78 453 patients" and then "were registered as the principal diagnoses in 76.6 % of all individual patients"? What does the first figure stand for? all patients with "any disease"? what does the percentage stand for?

-The authors stated at the end of this section: "Various genetic and social factors may influence the regional distribution of patients with Parkinsonism, as was already reported in the eastern part of our country. [13]". This should be rather put in the discussion section and better explained.

-In the sentence: " The ratio of the parkinsonian subtype of PSP (PSP-P) to the cerebellar subtype is 2:1 to 4:1 in 66 most countries. [15, 16]", the authors are rather talking about MSA and not PSP. This should be amended.

-Other minor changes suggested:

• The quality of figures and tables should be

Reviewer #2: In this manuscript, the authors attempted to determine epidemiological features of parkinsonian syndromes in the Czech Republic by a retrospective analysis of data available from the NHIS and the NRRHS. Further studies, such this one, on epidemiology of parkinsonism especially atypical parkinsonian syndromes are needed. However the authors should clarify some points, as follows:

1-The aim/objectives of the study is not well stated in the abstract and later in the introduction.

2- The authors should explain the exact meaning of the different codes used in the methodology part and precise all items/parameters that will be later detailed in the results

3-The results are not well defined both in the abstract and in the manuscript

4-The sentence "Various genetic and social factors may influence the regional distribution of patients with Parkinsonism, as was already reported in the eastern part of our country. [13]" should be in the discussion section.

5-The quality of figures and tables could be improved.

6. PLOS authors have the option to publish the peer review history of their article (what does this mean?). If published, this will include your full peer review and any attached files.

Reviewer #1: No

Reviewer #2: No

---

## [Author Response · Author response to Decision Letter 0]

10 Dec 2020

The aim of the study is not well stated. Reviewer #1 The aim was stated: 

Aim of the analyses: The main aim of our analyses was to identify the number of reported patients with Parkinson's disease (PD) – dg. G-20 (ICD-10) from the national health registries.

The results are not well defined (what do the authors mean by "We identified 78 453 patients? this figures stands for what exactly? Reviewer #1 Accepted – Thank you so much for this comment, we clarified it.

The conclusion part of the abstract included data not previously presented in the results. This should be addressed. Reviewer #1 Accepted – The data were included in the results section. 

Introduction:

The aim at the end of the introduction part is not clearly stated" to elucidate a number of patients treated for PD in the Czech Republic". The authors should be more specific. They stated later in the Materials and Methods part: "We analyzed epidemiological data intending to determine the regional and specific prevalence of Parkinsonism in the Czech Republic." Objectives should be mentioned all together at the end of the Introduction part. Reviewer #1 The clearly stated aim of the study was added as an extra section between “Introduction” and “Materials and Methods”. 

Materials and Methods:

The authors should define the exact meaning of the different codes used in this part: "The International Classification of Diseases 33 (ICD-10) diagnoses of G20 (Parkinson’s disease) and G23.1, G23.2, G23.3 (other degenerative disorders of basal ganglia), and G31.8 (another degenerative disease of basal ganglia)" Reviewer #1 Accepted – the data were analysed based on the record and codes reported by clinicians to the National health registries.

How do the authors explain the fact the diagnosis G23.3 (according to ICD-10) was not found in the analyzed data? and why did they decide to excluded this diagnosis from further evaluation? This is a result they could discuss later on in the discussion part. Reviewer #1 Accepted – the diagnosis G23.3 (according to ICD-10) was not found in the analyzed data – there were no record of the dg. G23.3 in the whole analysed dataset. The possible reason is, that there were no incidence of patients with dg. G23.3 which could relate to the problem that the clinicians are unable to diagnose it appropriately. Also published literature is confirming that the dg. G23.3 is not reported, the commonly used codes are 23.1, 23.2 and 23.9 (Harding, Z. 2019 https://journals.plos.org/plosone/article/file?id=10.1371/journal.pone.0198736&type=printable). 

Another possibility is that the clinicians do not show dg. multisystem atrophies correctly under the code G 23.3, but under the general one, for example G31.8.

What do the authors mean by "regional" and "specific" prevalence of Parkinsonism. This should be better specified in the Methods part. Reviewer #1 Accepted - there is no change in the methodology of data extraction. The regional analyses have been done based on the place where the care was provided – we have added information to the discussion section.

What do the authors mean by: "Atypical parkinsonian syndromes were 36 included due to overlap of clinical signs, at least during the initial stages of Parkinsonism."? How does this affect their methodology? this should be better explained. Reviewer #1 Accepted – there was no effect on the methodology – as we have included all the diagnoses mentioned in the methodology section. When we tried to identify only G20 – the data would not be accurate – it is based on the author’s clinical experiences with recording in the Czech Republic.

The authors should precise all the data analyzed later in the results in the methodology part. Reviewer #1 Accepted – double checked.

Results:

What do the authors mean by "we identified 78 453 patients" and then "were registered as the principal diagnoses in 76.6 % of all individual patients"? What does the first figure stand for? all patients with "any disease"? what does the percentage stand for? Reviewer #1 Accepted - The information was more precisely explained: We identified 205 490 records of patients with PD from national registries in the period from 2012 to 2018 (averaging 29 000 patients per year, shown in Fig.1), which means 78 453 unique patients in total (each patient could be recorded just one time during the year, but several times during the whole period from 2012 to 2018). We have found a growing number of records coded with ICD-10 of dg. G20, G23.1, G23.2, or G31.8 (N = 27 891 in 2012, and N = 30 612 in 2018).

The authors stated at the end of this section: "Various genetic and social factors may influence the regional distribution of patients with Parkinsonism, as was already reported in the eastern part of our country. [13]". This should be rather put in the discussion section and better explained. Reviewer #1 Accepted - we add the information in the discussion section.

In the sentence: "The ratio of the parkinsonian subtype of PSP (PSP-P) to the cerebellar subtype is 2:1 to 4:1 in 66 most countries. [15, 16]", the authors are rather talking about MSA and not PSP. This should be amended. Reviewer #1 Accepted - sorry for the misinterpretation – the text was changed. 

Other minor changes suggested:

• The quality of figures and tables should be Reviewer #1 We could not address this comment – maybe the comment was not finished?

The aim/objectives of the study is not well stated in the abstract and later in the introduction. Reviewer #2 The aim was stated: Aim of the analyses: The demographic description of reported patients with Parkinson's disease according to the International Classification of Diseases (ICD-10) from the national health registries. (in shortened version also in abstract).

The authors should explain the exact meaning of the different codes used in the methodology part and precise all items/parameters that will be later detailed in the results Reviewer #2 Accepted – double checked and added explanations.

The results are not well defined both in the abstract and in the manuscript Reviewer #2 Accepted – double checked and added explanations.

The sentence "Various genetic and social factors may influence the regional distribution of patients with Parkinsonism, as was already reported in the eastern part of our country. [13]" should be in the discussion section. Reviewer #2 Accepted - we add the information in the discussion section.

The quality of figures and tables could be improved. Reviewer #2 The figures are used from the data analyses system – we have no possibility to improve the quality.

Please ensure that your manuscript meets PLOS ONE's style requirements, including those for file naming. Editor Accepted – Thank you so much.

d) If you did not receive any funding for this study, please state: “The authors received no specific funding for this work.” Editor The authors received no specific funding for this work.

We note that you have indicated that data from this study are available upon request. PLOS only allows data to be available upon request if there are legal or ethical restrictions on sharing data publicly. For more information on unacceptable data access restrictions, please see http://journals.plos.org/plosone/s/data-availability#loc-unacceptable-data-access-restrictions.

We will update your Data Availability statement on your behalf to reflect the information you provide. Editor There is legal restrictions on sharing data publicly – all the data form the National registries must be kept safely and authors are working with the anonymised data. For the data you can ask the head of the department of data Analyses team from the Institute of Health Information and Statistics of the Czech Republic - Dr. Jiří Jarkovský (email: Jiri.Jarkovsky@uzis.cz)

We note that [Figure(s) 3] in your submission contain map images which may be copyrighted. All PLOS content is published under the Creative Commons Attribution License (CC BY 4.0), which means that the manuscript, images, and Supporting Information files will be freely available online, and any third party is permitted to access, download, copy, distribute, and use these materials in any way, even commercially, with proper attribution. For these reasons, we cannot publish previously copyrighted maps or satellite images created using proprietary data, such as Google software (Google Maps, Street View, and Earth). For more information, see our copyright guidelines: http://journals.plos.org/plosone/s/licenses-and-copyright.

1. You may seek permission from the original copyright holder of Figure(s) [3] to publish the content specifically under the CC BY 4.0 license. 

If you are unable to obtain permission from the original copyright holder to publish these figures under the CC BY 4.0 license or if the copyright holder’s requirements are incompatible with the CC BY 4.0 license, please either i) remove the figure or ii) supply a replacement figure that complies with the CC BY 4.0 license. Please check copyright information on all replacement figures and update the figure caption with source information. If applicable, please specify in the figure caption text when a figure is similar but not identical to the original image and is therefore for illustrative purposes only.

The following resources for replacing copyrighted map figures may be helpful... Editor The figure no. 3 was produced by our author’s team. We have published this kind of figure with different type of health data has been already published without any need for copyrighted.

---

## [Decision Letter · Decision Letter 1]

30 Dec 2020

PONE-D-20-31004R1

Representation of Parkinson's disease and atypical Parkinson's syndromes in the Czech Republic - A nationwide retrospective study

PLOS ONE

Dear Dr. Búřil,

Thank you for submitting your manuscript to PLOS ONE. After careful consideration, we feel that it has merit but does not fully meet PLOS ONE’s publication criteria as it currently stands. Therefore, we invite you to submit a revised version of the manuscript that addresses the points raised during the review process.

We look forward to receiving your revised manuscript.

Kind regards,

Weidong Le

Academic Editor

PLOS ONE

Reviewers' comments:

Reviewer's Responses to Questions

**Comments to the Author**

1. If the authors have adequately addressed your comments raised in a previous round of review and you feel that this manuscript is now acceptable for publication, you may indicate that here to bypass the “Comments to the Author” section, enter your conflict of interest statement in the “Confidential to Editor” section, and submit your "Accept" recommendation.

Reviewer #1: (No Response)

Reviewer #2: All comments have been addressed

2. Is the manuscript technically sound, and do the data support the conclusions?

Reviewer #1: No

Reviewer #2: Yes

3. Has the statistical analysis been performed appropriately and rigorously? 

Reviewer #1: Yes

Reviewer #2: Yes

4. Have the authors made all data underlying the findings in their manuscript fully available?

Reviewer #1: Yes

Reviewer #2: Yes

5. Is the manuscript presented in an intelligible fashion and written in standard English?

Reviewer #1: Yes

Reviewer #2: Yes

6. Review Comments to the Author

Reviewer #1: Most of the required modifications were amended. However, minor revisions are still needed:

1-In the abstract and in the manuscript: in the aim of the study, " The demographic description of reported patients with Parkinson's disease"==> The demographic description of reported patients with parkinsonism (including Parkinson's disease and atypical parkinsonian syndromes): the aim should be in accordance with the article title.

2-The authors should precise all the data analyzed later in the results in the methodology part. This was not amended.

3-The quality of figures and tables should be improved.

Reviewer #2: (No Response)

7. PLOS authors have the option to publish the peer review history of their article (what does this mean?). If published, this will include your full peer review and any attached files.

Reviewer #1: No

Reviewer #2: No

---

## [Author Response · Author response to Decision Letter 1]

6 Jan 2021

In the abstract and in the manuscript: in the aim of the study, " The demographic description of reported patients with Parkinson's disease"==> The demographic description of reported patients with parkinsonism (including Parkinson's disease and atypical parkinsonian syndromes): the aim should be in accordance with the article title. 

Accepted – the aim has been reworded according the reviewer´s comment.

The authors should precise all the data analyzed later in the results in the methodology part. This was not amended. 

Accepted – the information in the methodology part has been added.

The quality of figures and tables should be improved. 

Accepted – the figures were graphically edited and uploaded.

---

## [Decision Letter · Decision Letter 2]

18 Jan 2021

Representation of Parkinson's disease and atypical Parkinson's syndromes in the Czech Republic - A nationwide retrospective study

PONE-D-20-31004R2

Dear Dr. Búřil,

We’re pleased to inform you that your manuscript has been judged scientifically suitable for publication and will be formally accepted for publication once it meets all outstanding technical requirements.

Kind regards,

Weidong Le

Academic Editor

PLOS ONE

Additional Editor Comments (optional):

Reviewers' comments:

Reviewer's Responses to Questions

**Comments to the Author**

1. If the authors have adequately addressed your comments raised in a previous round of review and you feel that this manuscript is now acceptable for publication, you may indicate that here to bypass the “Comments to the Author” section, enter your conflict of interest statement in the “Confidential to Editor” section, and submit your "Accept" recommendation.

Reviewer #1: All comments have been addressed

2. Is the manuscript technically sound, and do the data support the conclusions?

Reviewer #1: Yes

3. Has the statistical analysis been performed appropriately and rigorously? 

Reviewer #1: Yes

4. Have the authors made all data underlying the findings in their manuscript fully available?

Reviewer #1: Yes

5. Is the manuscript presented in an intelligible fashion and written in standard English?

Reviewer #1: Yes

6. Review Comments to the Author

Reviewer #1: (No Response)

7. PLOS authors have the option to publish the peer review history of their article (what does this mean?). If published, this will include your full peer review and any attached files.

Reviewer #1: No

---

## [Editor Report · Acceptance letter]

21 Jan 2021

PONE-D-20-31004R2 

Representation of Parkinson's disease and atypical Parkinson's syndromes
in the Czech Republic - A nationwide retrospective study 

Dear Dr. Búřil:

I'm pleased to inform you that your manuscript has been deemed suitable for publication in PLOS ONE. Congratulations! Your manuscript is now with our production department. 

Kind regards, 

on behalf of

Dr. Weidong Le 

Academic Editor

PLOS ONE